

# Research progress on the effects of postharvest storage methods on melon quality

Haofei Wang[1,2,*], Jiayi Cui[1,2,*], Rui Bao[1,2], Hui Zhang[1,2], Zi Zhao[1,2], Xuanye Chen[1,2], Zhangfei Wu[3] and Chaonan Wang[1,2]

[1] Xinjiang Agricultural University, Xinjiang Special Melon and Fruit Variety Improvement and Logistics Transportation Joint Research Center, Urumqi, China
[2] Xinjiang Agricultural University, College of Horticulture, Urumqi, China
[3] Langfang Normal University, College of Life Science, Langfang, China
[*] These authors contributed equally to this work.

## ABSTRACT

**Background**. As an important global agricultural cash crop, melon has a long history of cultivation and a wide planting area. The physiological metabolism of melon after harvest is relatively strong; if not properly stored, melon is easily invaded by external pathogens during transportation, resulting in economic losses and greatly limiting its production, development and market supply. Therefore, the storage and freshness of melon are the main challenges in realizing the annual supply of melon, so postharvest storage has received increasing amounts of attention from researchers.

**Methods**. This study used academic, PubMed, and Web of Science resources to retrieve keywords related to postharvest storage and melon quality; read, refined, classified, and sorted the retrieved literature; sorted and summarized the relevant research results; and finally completed this article.

**Results**. This article reviews the mechanism and effects of physical, chemical and biological preservation techniques on the sensory quality, compound contents and respiratory physiological activities of different varieties of melon fruits. When maintaining normal metabolism and not producing physiological disorders, melon inhibits cell wall metabolism, reactive oxygen species metabolism and the ethylene biosynthesis pathway, *etc*., to the greatest extent during postharvest storage, thereby reducing the material consumption of fruits, delaying the ripening and senescence process, and prolonging the postharvest life and shelf life.

**Conclusion**. The literature provides a theoretical basis for postharvest preservation technology in the melon industry in the future and provides corresponding guidance for the development of the melon industry.

Corresponding author
Chaonan Wang, 411803424@qq.com

# INTRODUCTION

Melon (*Cucumis melo* L.) is one of the most important cucurbitaceous crops worldwide and occupies approximately 3% of the cultivated area used for all types of vegetables.

Melon flesh is thick and juicy (accounting for at least 65% of the total weight) and is rich in various nutrients, such as carbohydrates, citric acid, carotene and vitamins (*Fundo et al., 2017*). Melons have a long history of cultivation in China. As the largest producer and consumer of melons, China produced approximately 140.716 million tons of melons in 2021 (*Food and Agriculture Organization of the United Nations, 2024*).

Melon is a typical respiratory jump fruit. During postharvest storage, a series of physiological and biochemical reactions occur inside the fruit, including changes in color, size, taste, flavor, texture, *etc*. These important indicators determine fruit quality and marketability (*Farcuh et al., 2020a*; *Farcuh et al., 2020b*). Often, many breeding programs will focus more on improving shelf life at the expense of flavor to improve marketability, so preserving the original flavor and quality of food while improving product competitiveness through research into postharvest technologies is key to meeting consumer demand and maintaining lasting profitability (*Klee & Tieman, 2018*; *Farcuh et al., 2020a*; *Farcuh et al., 2020b*). Domestic melon varieties have a high yield, but their preservation processes may be relatively primitive compared with those of other fruits, affecting their original flavor, nutrition and texture, which seriously restricts their production and sales. This causes economic losses for farmers (*Xu et al., 2019*).

Therefore, this article summarizes the effects of different treatments on the sensory quality, compound content, and respiratory physiological activity of melon fruits, which is highly important for the development of pollution-free melon production, the acceleration of postharvest treatment and processing technology research, the extension of the postharvest storage industry chain of melon, and the provision of theoretical and practical guidance and support for the sustainable development of the melon industry.

## SURVEY METHODOLOGY

In this article, the keywords related to the postharvest treatment and storage and preservation of different varieties of melon were retrieved from academic, PubMed and Web of Science resources, and the specific quality and physiological changes corresponding to different postharvest storage methods of melon were analyzed from the internal and external quality of melon fruits. The refinement direction of respiratory regulation and the specific quality and physiological changes of melon corresponding to different postharvest storage methods were analyzed. In this process, many retrieved studies were read, refined, classified and sorted. Finally, this findings were summarized and are presented in this article.

### Effects of different storage methods on the sensory quality of melon fruits

#### Influence on melon firmness

Firmness is an indicator that reflects the quality changes and degree of softening of fruits during postharvest storage (*Ahmad et al., 2023*) and is closely related to the cell wall (CW) structure (Fig. 1). Cell walls are mainly composed of pectin, cellulose, hemicellulose, and glycoproteins (*Uluisik & Seymour, 2020*). CWs are present in all plants and provide mechanical strength by acting as an exoskeleton, regulating osmotic pressure, and

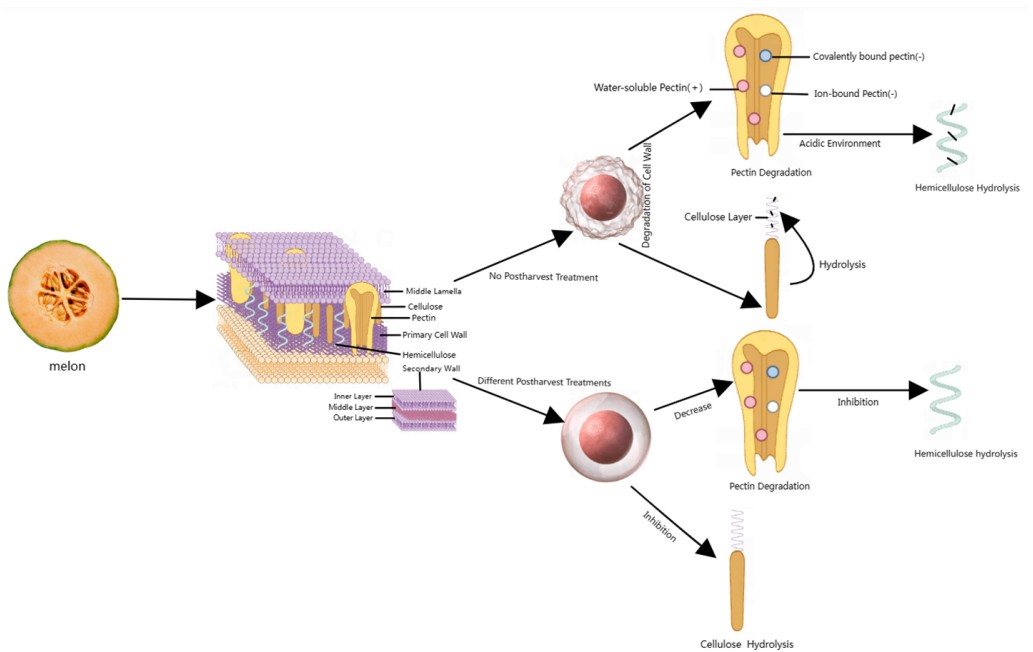

**Figure 1** **Corresponding metabolic processes of cell wall substances under different postharvest treated and untreated conditions.** The plant cell wall is roughly represented as middle lamella, primary cell wall, secondary wall. Red spheres represent cell wall models. (+) represents an increase, (-) represents a decrease. Made in Medpeer (https://image.medpeer.cn/show/index/template).

controlling cell adhesion during cell expansion. In addition, CWs function as a diffusion barrier, protecting and preventing plasma membrane disruption and acting as a barrier against pathogens and herbivores.

The main reason for fruit softening is the destruction of the CWs (*Pott, Vallarino & Osorio, 2020*). As the product organs mature, the network structure of CWs continuously disintegrates, and the original pectin in the CWs gradually transforms into soluble pectin. The CW structure becomes loose, and ultimately, the firmness decreases (*Posé et al., 2019*). During storage, the firmness of melon fruits decreases with the progression of fruit ripening. Firmness is strongly correlated with sugar content, and the acoustic response of melon fruits is an important parameter for estimating firmness, soluble solid content (SSC), grade and internal quality (*Sun et al., 2010*).

Softening is a complex process often associated with the degradation of fruit cell wall components and involves an increase in water-soluble pectin and a decrease in insoluble and covalently bound pectin (*Yan et al., 2023*). According to *Le Nguyen et al*'s (*2019*) detection of acoustic vibrations in melon fruits, conventional gaseous 1-methylcyclopropylene (1-MCP) fumigation and liquid 1-MCP microbubble treatment for 20 to 30 min can delay the softening of 'Donatello' melons. *Ergun et al. (2005)* reported that during storage at 20 °C, 1.5 µL/L 1-MCP gas treatment of 'Galia' melon for 24 h delayed fruit softening, extended the shelf life of the yellow harvest to 11 days, and extended the shelf life of the green harvest to 20 days.

Treating melon fruits with an aqueous calcium solution before or after harvesting could increase the content of calcium ions in cells and improve the stability of pectin (*Silveir et al., 2011*). *Zhang et al. (2022b)* studied the 'Golden Queen' hami melon, which was treated with a 1.5% chitosan solution for 10 min before refrigeration at 3 °C, and the results revealed that chitosan treatment significantly reduced fruit softening. *Hatami et al. (2019)* harvested Dudam melon 21 days after flowering, and long-term storage at a low temperature of 13 °C was conducive to maintaining its firmness. *Tappi et al. (2016)* used empty dipping (VI) as a processing operation, which can better maintain the texture of processed melons during storage after VI treatment.

Ethylene treatment significantly accelerates the decomposition of cell wall membranes (CWMs) and covalently bound pectin (CSP), promotes an increase in the water-soluble pectin (WSP) content, accelerates the degradation of hemicellulose and cellulose, promotes the decomposition of ion-bound pectin (ISP) at the later stage of storage, and accelerates the decrease in fruit firmness. During storage, treatment with the ethyl synthesis inhibitor (AVG) 1-MCP and low temperatures can significantly delay the degradation of the CWM and CSP components and inhibit the increase in the WSP content. It is beneficial for maintaining the stable content of ISP, hemicellulose and cellulose; reducing the decomposition rate; and helping to maintain the firmness of fruit (*Zhang et al., 2022a*; *Zhang et al., 2022b*).

### Influence on melon weight loss rate and decay rate

Melon fruits are divided into respiratory climacteric and nonrespiratory climacteric types. During postharvest storage, water is lost from the fruit tissue of respiratory climacteric sweet melons, and the activities of various hydrolytic and respiratory enzymes increase sharply. Fruit respiration is accelerated, and related nutrients are decomposed, thus increasing the weight loss rate of the fruit. *Zhang et al.* (*2022a*; *2022b*) studied three Hami melon varieties stored at $20 \pm 1$ °C and 25% relative humidity (RH) for 18 days. The weight loss results indicated that the storage quality of 'Chougua' was obviously better than that of 'Xizhoumi 17' and 'Jinhuami 25'. The application of rapeseed meal enzymatic hydrolysate as a natural preservative in postharvest storage and the preservation of cantaloupe can effectively reduce the quality loss rate of cantaloupe (*Wang et al., 2017*). *Kim et al. (2010)* reported that a low temperature ranging from 7 to 10 °C and a high RH ranging from 90 to 95% are suitable environmental conditions for delaying weight loss in fresh oriental melon fruits. Controlled atmosphere (CA) storage or modified atmosphere (MA) packaging can be used as supplemental treatments. Research has shown that, compared to separate treatments of 0.75 g/L chitosan and 0.05 mL/kg ethanol, composite treatment further delays the loss of melon quality (*Sun et al., 2023*). Moreover, experiments have shown that the combination of postharvest chitosan coating and low-temperature (5 °C) storage treatment has a more prominent effect on delaying the rate of fruit weight loss (*Wang et al., 2016*).

The decay index is an important index of the effect of fruit storage and preservation (*Zhang et al., 2023a*; *Zhang et al., 2023b*; *Zhang et al., 2023c*). A previous study revealed that when melons were stored at $20 \pm 1$ °C and 25% RH for 18 days, compared with those of 'Xizhoumi 17' and 'Jinhuami 25', the decay rate of 'Chougua' significantly decreased in

the later stage of storage (*Zhang et al., 2022a*; *Zhang et al., 2022b*). *Gal et al. (2006)* stored Galia melon for 15 days at 5 °C and for 3 days at 20 °C. The results showed that the optimal treatment with 1-MCP for suppressing the ripening of fruits harvested at the green/yellow stage of maturity was 300 nl l$^{-1}$ for 24 h at 20 °C, and the decay rate of fruits after treatment was significantly lower than that of the control group. However, it has been also reported that disease severity can be severe in both control and 1-MCP-treated melons (*Sun et al., 2010*). *Hoberg et al. (2003)* conducted postharvest and end-of-year storage and marketing simulations. After 16 days of storage at 5 °C and an additional 3 days at 20 °C, only cultivar 'C-8' had a poor general appearance due to its relatively high decay incidence compared to those of the cultivars '5080', 'Ideal' and '7302'.

## Influence on melon color indices

The color and luster of fruits are the most intuitive manifestations during storage and are important indicators that affect their commercial value (*Afonso et al., 2023*). Many studies have been conducted on the preservation of melon fruit color. *Du et al. (2016)* used 'Xizhou Mi 25' as the test material and cooled it with 0 °C ice water for 120 min and then refrigerated it at 6 °C for 6 h after harvest, which slowed the color change during storage and effectively maintained the color of Hami melons. Among the two low-temperature treatment conditions of 10 °C and 15 °C, the 10 °C treatment can effectively delay the color change in melon fruit flesh, maintain fruit brightness, and maintain the original color of the fruit (*Shao, 2019*). Yao et al. used the melon variety 'Xizhou Mi 25' as the test material and continuously irradiated it with red, blue, and white monochromatic weak light (30 lx) using light-emitting diodes at 7 °C. Red light has the greatest effect on maintaining the color of melon skin (*Yao et al., 2023*). In the study of *Agehara et al. (2018)*, 1-MCP immersion had a small effect on the initial color transition from green to light yellow, but it delayed the subsequent development to orange, extending the fruit shelf life with a desirable ripe skin color. *Watkins (2006)* confirmed that 1-MCP treatment could inhibit the synthesis and accumulation of pigments and delay the color transformation of fruits. *Jin et al. (2013)* studied the effects of ethanol steam treatment (0.5 mL/kg and 3 mL/kg) at 23 °C on the postharvest storage quality of 'Oriental' melon fruits. The effects of ethanol vapor treatment on skin color intensity and color change in melons were dependent on the dose of ethanol, with a better appearance of melons at 0.5 mL/kg than at 3 mL/kg. The use of flusilazole fungicide to treat fruits results in greater changes in the total color difference ΔE during storage (*Du et al., 2015*).

## Effects of different storage methods on compound contents in melon
### Influence on melon soluble solids content

The soluble solids content (SSC) is the main characteristic indicator for evaluating the flavor and nutritional status of melon fruits and can directly reflect the maturity and quality status of the fruit (*Yao et al., 2023*). Soluble solids mainly refer to the soluble sugars, amino acids, vitamins, and minerals contained in the cell fluid, among which sugars are the main type. Cotreatment with 1-methylcyclopropene (1-MCP) and enhanced freshness formulation (EFF) can delay the trend of decreasing SSC in fruits (*Bai et al., 2014a*; *Bai et al., 2014b*). Furthermore, in another study, Caldeo was stored at 6 °C for 13 days, the fruits

were refrigerated and treated with 0.15 ppm gaseous ozone during the day and 0.3 ppm at night, and the analysis showed that the control fruit (CK) stored at a normal temperature could effectively inhibit the increase in total soluble solids (TSS) in fruit and improve the quality of melon (*Toti, Carboni & Botondi, 2018*). Zhang et al. studied the 'Golden Queen' Hami melon, which was treated with a 1.5% chitosan solution for 10 min before being refrigerated at 3 °C. Another study revealed that chitosan not only retarded the loss of SSC and fruit ripening in melon but also reduced fruit softening and chilling injury (*Zhang et al., 2022a*; *Zhang et al., 2022b*). Research has shown that heat treatment can effectively delay the decrease in the TSS content in the fruits of Jinmi 3 and Jiashi Gua fruits of three types of melon materials with similar TSS contents in the early stage of storage (*Yin et al., 2022*). 'Xizhou Mi 25' was as the test material in a different study, and the plants were soaked in 1.0% SA for 5 min after harvest and stored in cold storage at 6−8 °C. The effect on the SSC of the fruit pulp was not significant (*Yao et al., 2018*). The 0 (CK), 50 µL/L, 100 µL/L, and 200 µL/L immersion treatments were used in their study.

## Effects on the vitamin C content in melon

Vitamin C (Vc) is an important nutrient in fruits that contributes to the antioxidant capacity of plant cells by detoxifying reactive oxygen species (ROS) and free radicals (*Rey, Zacarías & Rodrigo, 2020*). Research has shown that temperature affects the Vc content in fruits, and as temperature decreases, the Vc content increases. *Yousuf, Srivastava & Ahmad (2020)* analyzed fresh-cut melons treated with different concentrations of lemon extract (0, 5, 10 and 15%) and coated them with 0 and 5% soy protein isolate (SPI), and the results showed that these treatments were effective at retaining vitamin C in melon samples (*Kim et al., 2015*). Research has shown that when different gas ratios ($O_2$:$CO_2$) are used to treat 'Xizhou Mi 25' Hami melons, the rate of change in the Vc content in the 6% $CO_2$+1% $CO_2$ and 3% $O_2$+1% $CO_2$ treatments is slower than that in the control group (*Yousuf, Srivastava & Ahmad, 2020*). *Wang et al. (2023)* used three different concentrations of melatonin to maintain the content of ascorbic acid, and the fruit treated with 0.5 mmol/L MT showed the greatest effect.

## Influence on the titratable acidity content of melon

The titratable acid (TA) content is an important indicator used to reflect the flesh quality of fruits and has a significant impact on the intrinsic quality of fruits (*Zhang et al., 2023a*; *Zhang et al., 2023b*; *Zhang et al., 2023c*). Titrable acidity can serve as a substrate for respiration, and as storage time progresses, the titratable acidity gradually decreases, leading to a decrease in its content and quality. The main organic acid in cantaloupe is citric acid. Research has shown that 2 µL/L1-MCP fumigation for 24 h can be used to treat 'Berserksin' cantaloupe, which is stored in cold storage at 7 °C (85∼90% RH), which significantly inhibits the decrease in titratable acid content (*Zhang, 2018*). Moreover, 1-MCP treatment of 'Xizhou Mi 25' under storage at 5 °C effectively inhibited the titratable acid content of the fruit during postharvest storage of sweet melons. A study was conducted on the physiological changes and quality of 'Jiashigua' fruit during storage using eight different modified atmosphere treatments. A gas ratio of 7% $CO_2$+5% $O_2$+88% $N_2$ was shown to

better maintain the titratable acid content of the fruit (*Zhang et al., 2011*). After treating Jinxiangyu melon with hot water at 53 °C, the decrease in the titratable acid content has been found to be delayed, indicating that hot water treatment can better maintain the storage quality of the fruit (*Zhang et al., 2010*). *Wang et al. (2012)* reported the slowest decrease in titratable acid in fruits treated with 1-MCP combined with $Na_2S_2O_5$.

## Effects on enzyme activities related to postharvest metabolism in melon

Polygalacturonase (PG) is an important hydrolase that breaks down pectin molecules by hydrolyzing 1,4- $\alpha$-D-galactoside bonds in polygalacturonic acid in the CWs, thus disintegrating the CW structure and softening the fruit (*Goukh & Elhassan, 2017*). Pectin methylesterase (PME) is involved mainly in the demethylation of pectin substances in the cell walls and can remove methyl groups from pectin in the cell walls, thus accelerating the degradation of PG metabolism  (Fig. 2) (*Yu et al., 2023*). Studies have shown that calcium treatment can effectively inhibit the degradation of cell wall substances (*Liu et al., 2023a*; *Liu et al., 2023b*). Moreover, PME demethylation products are precursor substrates that significantly inhibit the activities of enzymes related to fruit softening, such as PG, PME and $\beta$-glucosidase (*Jeong et al., 2004*). Under conditions of 20 $\pm1$ °C and 25% RH, PME deesterification can be reduced by inhibiting PME activity in fruits, thus inhibiting the hydrolysis of the polygalacturonic acid main chain catalyzed by PG. In a study of thick skin melons, soaking melon in hot water at 53 °C for 3 min effectively inhibits the activities of polygalacturonase, pectinesterase, $\beta$-galactosidase and the endonuclide 1,4- $\beta$-D-glucanase; maintains the firmness of the fruit during storage; and increases the contents of suberin and callose (*Yuan et al., 2013*).

Peroxidase (POD) is an important oxidoreductase in fruit that can catalyze $H_2O_2$ to oxidize phenolic substances to produce quinones, and the quinones are further concentrated to form brown substances (*Luo, Chen & Xie, 2011*). Under postharvest conditions, fruits usually receive multiple signals at the same time, and ROS are easily destroyed (*Karlia, Yonadita & Sony, 2020*). Superoxide dismutase (SOD) and catalase (CAT), two major antioxidant enzymes, can effectively control the imbalance of ROS metabolism, thereby inhibiting oxidative stress during storage  (Fig. 3) (*Chen et al., 2015*).

1-MCP treatment can effectively maintain the activities of SOD, ascorbate oxidase (APX) and POD and improve the activity of CAT in muskmelon during the middle stage of storage (4~5 days). However, ethylene treatment inhibits the activity of reactive oxygen-scavenging enzymes in fruits, which diminishes the scavenging of reactive oxygen radicals. The aggregation of ROS leads to intracellular oxidative stress, which includes the peroxidation of unsaturated fatty acids. Peroxidation triggers a free radical chain reaction that disrupts the lipid structure of cell membranes, leading to cellular senescence and tissue aging (*Zhang, 2018*). *Yuan et al. (2013)* soaked melon in hot water at 53 °C for 3 min, and the results showed that the activity of POD significantly increased. Studies have shown that oxalic acid treatment can reduce the accumulation of ROS by stimulating the antioxidant system of melons. Among them, enzymes, such as SOD, POD and CAT, are important antioxidant enzymes in plant cells that can help to scavenge reactive oxygen radicals and

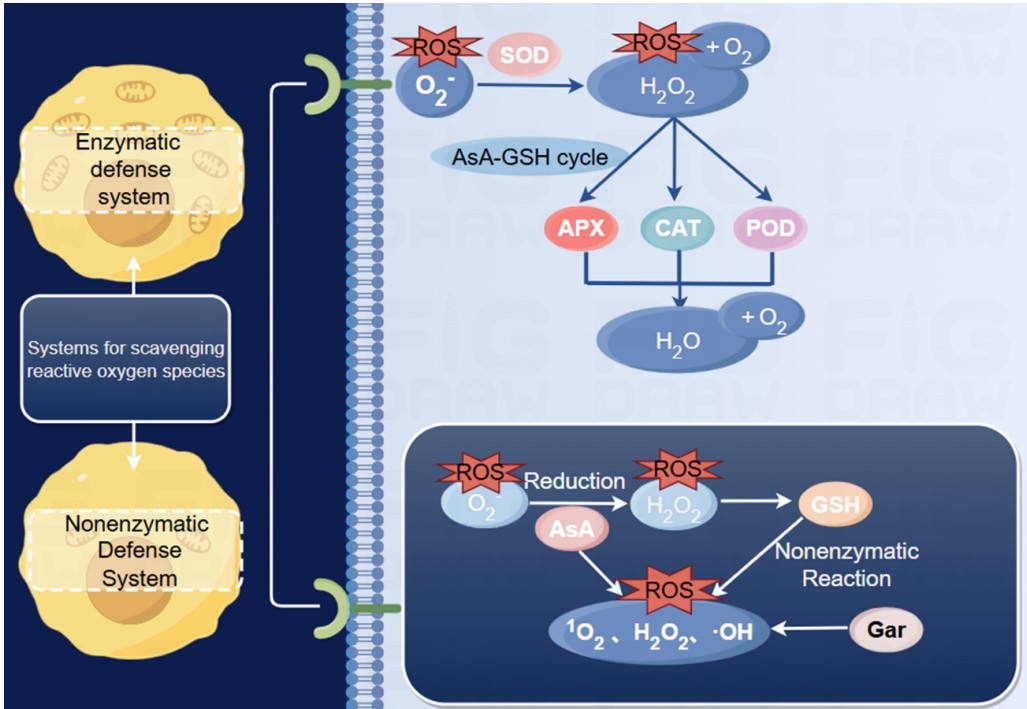

**Figure 2** Diagram of the reaction process of fruit cell wall metabolism-related enzymes to pectin during ripening and softening (*Kohli, Kalia & Gupta, 2015*; *Zhang et al., 2022a*; *Zhang et al., 2022b*). The plant cell wall is roughly represented as middle lamella, primary cell wall, secondary wall. Red spheres represent cell wall models. (+) represents an increase, (-) represents a decrease. Made in Figdraw (https://www.figdraw.com/static/index.html#/).

**Figure 3** Reactive oxygen metabolism reaction process (*Ren et al., 2012*; *Ventura-Aguilar et al., 2013*). Includes enzymatic and non-enzymatic defense systems. Superoxide dismutase catalyzes the disproportionation reaction of $O_2^-$ to produce $O_2$ and $H_2O_2$; Peroxidase, catalase, ascorbate oxidase are all involved in the removal of $H_2O_2$, but ascorbate oxidase needs to participate in the AsA-GSH circulatory system. Reduced ascorbic acid reduces $O_2^-$ to $H_2O_2$, eliminating various reactive oxygen species. On the other hand, GSH eliminates hydrogen peroxide, but also reacts non-enzymatically with many reactive oxygen species, and Gar effectively eliminates reactive oxygen species.The abbreviations for antioxidant enzymes are used in the figure: superoxide dismutase (SOD), Peroxidase (POD), catalase (CAT), ascorbate oxidase (APX); The abbreviations for antioxidants are used in the figure: reduced ascorbic acid (AsA), Glutathione (GSH), Carotene (Gar). Made in Figdraw (https://www.figdraw.com/static/index.html#/).

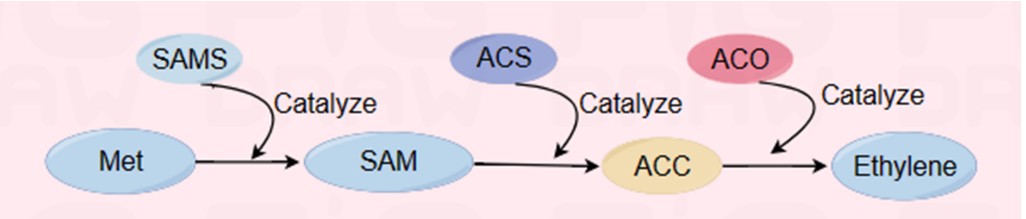

**Figure 4** **Ethylene biosynthesis reaction pathway** (*Bleecker & Kende, 2000*). Among them, ACC is an important intermediate, ACC synthase (ACS) and ACC oxidase (ACO) are the key enzymes regulating ethylene synthesis. The following abbreviations are used in the figure: methionine (Met), S-adenosy-L-methionine synthetase (SAMS), S-adenosylmethionine (SAM), ACC synthase (ACS), 1-aminocyclopropane-1-carboxylic acid (ACC), ACC oxidase (ACO). Made in Figdraw (https://www.figdraw.com/static/index.html#/).

maintain the intracellular redox balance. Melatonin treatment can simultaneously increase SOD, POD, CAT and APX activities, reduce $O_2^-$ and $H_2O_2$ levels, delay postharvest senescence, and maintain melon fruit quality (*Wang et al., 2018*).

Ethylene is a key factor affecting fruit ripening and senescence (*Lee et al., 2023a*; *Lee et al., 2023b*). The purpose of delaying and controlling the ripening and senescence process of melon fruits can be achieved by regulating the release of ethylene and the activities of the key enzymes ACS and ACO in the ethylene biosynthesis pathway (Fig. 4) (*Li et al., 2023*). Yazhong et al. reported the effects of ethanol vapor treatment (0.5 mL/kg and 3 mL/kg) on the postharvest storage of melon fruits. Both treatments significantly ($P < 0.05$) suppressed the internal ethylene concentration (IEC) during storage at 23 °C, which was evident from the decreases in 1-amino-cyclopropane-1-carboxylic acid (ACC) synthase (ACS) and ACC oxidase (ACO) activities and the inhibition of ACC biosynthesis (*Jin et al., 2013*); moreover, the effect of the 0.5 mL/kg treatment was greater than that of the 3 mL/kg treatment. According to Li et al., 1-MCP treatment led to a decrease in autocatalytic ethylene synthesis, mainly due to the inhibition of ACS and ACO activities, and the inhibition of fruit softening by 1-MCP may be related to a decrease in ethylene production (*Wang et al., 2017*). *Ayub et al. (1996)* showed that melons that express the antisense 1-amino-cyclopropane-1-carboxylic acid (ACC) oxidase gene were produced by genetic engineering preservation treatment, indicating that melons expressed by the ACC oxidase gene can inhibit postripening and senescence of fruits and prolong their shelf life.

**Effect on the membrane permeability of melon plants**

Malondialdehyde (MDA) is produced by lipid peroxidation in tissues or organs of plants due to senescence or damage under stress conditions. The MDA content is an important indicator of the degree of fruit senescence (*Khaliq et al., 2016*; *Carvalho et al., 2016*). *Wang et al. (2023)* treated melon plants with 0.5 mmol/L melatonin to effectively reduce the content of malondialdehyde, thus significantly delaying aging and prolonging storage life. Rokayya *Sami et al*'s. (*2021*) treatment of melon with nanomaterials delayed the increase in MDA, demonstrating that nanotreatment can inhibit lipid peroxidation during storage. When the 1-MCP compound fungicide imazole is used to treat the Golden Red Treasure

thick-skinned melon, the MDA content in the fruit remains low over time (*Yin et al., 2015*). The soaking of 'Yujinxiang' muskmelon with 4 mmol/L salicylic acid for 10 min induced an oxygen explosion in thick-skinned muskmelon fruits and inhibited MDA production (*Yang et al., 2021*). Compared with the initial storage values under three low-temperature (5 °C, 8 °C, and 12 °C) and room temperature conditions, the malondialdehyde content of melon stored at 5 °C for 28 days increases by approximately 1.5 times, significantly delaying the increase in malondialdehyde content (*Tian et al., 2022*).

## Effects of different storage methods on the physiological respiratory activities of melon fruits

### Effects on fruit respiratory intensity

During storage, an increase in postharvest fruit respiration may promote biochemical processes and thus reduce postharvest fruit quality (*Lu, Yang & Xue, 2023*). The combined treatment of 1.5% $CaCl_2$ and 1 μL/L 1-methylcyclopropylene in melon can effectively inhibit the respiratory intensity of melon fruits (*Li et al., 2023*). The use of hydrogen peroxide ($H_2O_2$) as a potential postharvest treatment for Jiashi muskmelon has been investigated. In *Chen et al. (2015)*, muskmelon fruits were treated with 3, 4 or 5% (v/v) $H_2O_2$, placed on shelves, and stored at $6 \pm 1$ °C and 80–90% RH for 60 days. Physiological responses, nutritional attributes and decay rates were evaluated. $H_2O_2$ treatment effectively inhibited respiration and ethylene production rates; delayed the decrease in firmness, soluble solid concentration (SSC) and mass loss; delayed the changes in the contents of Vc, titratable acidity, and total phenolic substances, *i.e.,* flavonoids and anthocyanin; and reduced postharvest decay in melons during storage .

Hot water treatment combined with O-carboxymethyl chitosan (CMC) coating has been shown to be more effective at preserving Hami melons, as indicated by decreased respiration rates (*Zhou et al., 2020*). Studies have shown that melon slices treated with calcium propionate, tartrate, lactate, ascorbate, or chlorine have a lower respiratory rate at the end of their shelf life (*Silveir et al., 2011*). The application of the compound preparation of chitosan and trans-cinnamaldehyde to melon fruits significantly inhibits the respiration of melon fruits, and the fruit quality significantly improves compared with that of the control group (*Carvalho et al., 2016*).

### Effects on postharvest ethylene release in melon

Ethylene, a plant hormone, is produced during fruit ripening and can affect the postripening senescence of fruits (*Zhang et al., 2019*). After 1 μL/L 1-MCP treatment and long-distance transport at room temperature, ethylene release in melon fruits is effectively inhibited. Ma et al. also proved that treatment with 1 μL/L 1-MCP could significantly inhibit the release of ethylene in fruits during storage and delay the duration of the ethylene peak (*Ayub et al., 1996*). 1-Methylcyclopropene microfoam (1-MCPMB) treatment for 20 min or 30 min can greatly reduce the amount of ethylene released from melon plants during storage.

According to the amount of ethylene released, the effect of 0.6 μL/L 1-MCPMB treatment for 20–30 min is the same as that of gaseous fumigation (*Le Nguyen et al., 2019*). Through genetic engineering, the antisense ACC oxidase gene has been shown to be expressed in melon, decreasing the ethylene content in the fruits to less than 1% of the total ethylene

content in the control group and significantly inhibiting the ripening of melon (*Ayub et al., 1996*). A study by *Zhang et al. (2019)* showed that the combination of 0.18 mol/L CaCl$_2$ and 1 µL/L 1-MCP had a better inhibitory effect on ethylene release, which significantly delayed the softening of melon and extended its shelf life.

### Effects on the aroma volatiles of melon

Aroma components can objectively reflect the maturity of a fruit; this maturity is an important factor determining the market value and consumer satisfaction of fresh melons and can be used to evaluate the postharvest quality of fruits (*Padilla-Jiménez et al., 2021*). The intermediate aroma components of melon fruits are mainly esters, alcohols and aldehydes. The combined treatment of chitosan and ethanol can delay decreases in fruit fragrance and flavor, such as decreases in alcohol and aldehyde contents, after storage for 12 days (*Sun, 2021*). The total contents of aroma volatile compounds, esters, alcohols and aldehydes in melon are greater than those under control and 1-MCP treatments. Moreover, the peak time of aroma volatilization in the cotreated melons is later than that in the control and 1-MCP treatments. Therefore, cotreatment with 1-MCP and a membrane degradation inhibitor is more beneficial for delaying ripening and senescence, maintaining fruit quality, and improving the level of volatile aroma compounds than 1-MCP treatment alone (*Bai et al., 2014a*; *Bai et al., 2014b*). Refrigeration can change the flavor of melon fruits. After refrigeration, the total ester content decreases, especially volatile acetate esters (VAEs), and consumers prefer more ester flavor types. Transcriptomic analysis has revealed that the transcriptional abundance of several flavor-related genes involved in the fatty acid and amino acid pathways of melons decreases after refrigeration. In *Zhang et al. (2023a)*; *Zhang et al. (2023b)*; *Zhang et al. (2023c)*, the Japanese sweet bag melon was stored at 6 °C by rapid and slow cooling methods. The alcohol and phenol contents in the fruits treated by the two cooling methods increased slowly, and the aldehyde content increased more. At the end of storage, the ester content of fruits in the slow-descending treatment group was greater than that in the fast-descending treatment group.

## DISCUSSION

The sensory quality of melons is a complex quality influenced by taste, texture and color. In particular, firmness and juiciness are the most important textural attributes for consumers in the market (*Toivonen & Brummell, 2008*). During ripening, the thin cell walls of the fruit gradually disintegrate, leading to changes in the mechanical properties of the cells. At the same time, dissolution of the thin middle layer reduces intercellular adhesion. These changes, together with the loss of expansion pressure, result in the fruit becoming soft and intolerant to storage (*Redgwell et al., 1997*). The process of cell wall breakdown involves the breakdown of matrix glycans, the solubilization and breakdown of pectin, and the loss of neutral sugars from pectin side chains (*Posé et al., 2019*). The impact of these changes on the application of different preservatives remains somewhat unclear and needs to be explored in further studies. The cell wall is the first barrier against pathogen invasion in higher plants, and according to the results of previous studies, there is a strong correlation between firmness and decay incidence (*Yuan et al., 2013*). The physiological

disorganization caused by the production of rot often has a direct effect on the rate of weight loss. Different preservatives may synthesize or degrade pigments (chlorophyll, carotenoids, anthocyanins, *etc.*) in melons during storage, thus affecting changes in fruit color development. The cell wall is the first barrier against pathogen invasion in higher plants, and according to the results of previous studies, there is a strong correlation between hardness and decay incidence (*Yuan et al., 2013*). The physiological disorganization caused by the production of rot often has a direct effect on the rate of weight loss. Different preservatives may synthesize or degrade pigments (chlorophyll, carotenoids, anthocyanins, *etc.*) in melons during storage, thus affecting changes in fruit color development.

Many previous studies have confirmed that different kinds of preservative treatments can effectively maintain the content of soluble solids and vitamin C and inhibit the decrease in the titratable acidity content of fruits. These effects may be related to the physiological processes and chemical reactions within the fruits, which may affect the quality characteristics of the fruits, and the sensitivity of the fruits to the different kinds of preservatives may result in differences in the effectiveness of the different preservatives on the maintenance of the quality indices of the fruits. During postharvest storage, excess ROS damages cell membranes, while low levels of ROS are scavenged by the antioxidant system (*Meitha, Pramesti & Suhandono, 2020*). The accumulation of ROS can activate the antioxidant system, leading to an increase in antioxidant enzymes and substances, which have been demonstrated to be involved in the mechanisms of lipid peroxidation and delayed senescence in many horticultural crop species, which, in turn, act to inhibit the accumulation of ROS (*Wang et al., 2023*). In contrast, the accumulation of MDA, the end product of lipid peroxidation, indicates the presence of oxidizing free radicals in cells, which disrupt the cell membrane structure and interfere with normal physiological metabolism (*Dong et al., 2021*). This may be related to the storage environment, oxygen concentration, intracellular enzyme activity, *etc.*, and balancing the production and elimination of reactive oxygen radicals plays an important role in fruit senescence.

In melon, respiratory intensity and ethylene release usually increase with fruit ripening. There may be a relationship between this increase and aroma volatiles. Little research has been done on this topic, and more in-depth exploration and experiments are needed to prove their interrelationship. Aroma is one of the most important factors influencing fruit quality and consumer preference (*Lucchetta, manriquez & EI-Sharkawy, 2007*) and is influenced by a range of attributes, such as variety, ethylene and ripening stage (*Jia, Arika & Okamoto, 2005*; *Li, Jia & Okamoto, 2007*). In melon, alcohols, aldehydes, and acetates are the main volatile components, and these volatiles determine the unique flavor of melon. However, with the gradual ripening and aging of the fruit, the content of alcohols and aldehydes decreases, while the content of esters increases. The major esters are acetate, followed by oxalate and other esters (*Bai et al., 2014a*; *Bai et al., 2014b*). Specific storage treatments may promote the release of certain aroma components or the formation of new aroma substances, increasing the melon aroma and depth of flavor. It is of great significance for the mining and enhancement of melon aroma in the food industry and agriculture.

## CONCLUSION AND FUTURE PERSPECTIVE

In this article, we reviewed related studies on the storage and quality preservation of melon fruits and found that different storage conditions strongly affect the quality and physiology of melon fruits. To date, some in-depth research has been conducted on the preservation technology of muskmelon. The methods used to store and preserve muskmelon after harvest mainly include chemical preservation, physical preservation and biological preservation. In the field of melon storage and preservation, an increasing number of researchers and producers tend to use more natural and healthy physical and biological methods to maintain the freshness and quality of the fruit. The application of 1-MCP in melon storage and preservation has become more promising, with a shift from single 1-MCP treatments to cotreatments, from conventional fumigation in gaseous form to shorter liquid forms, and more flexible and shorter liquid applications of 1-MCP when closed storage chambers are unavailable. This provides new ideas for the study of 1-MCP preservation in melons, and synergistic effects can be achieved by combining 1-MCP with other preservatives. However, there are some limitations and potential risks, and the dosage and concentration requirements are very strict. Too high or too low is not conducive to fruit storage and can even accelerate the deterioration of fruits and vegetables. Therefore, it is recommended to use 1-MCP in combination with appropriate detergents to inhibit melon decay, and it is possible to study the preservative mechanism of 1-MCP treatment on melons during the storage period *via* microbial sequencing.

In the process of using different preservatives, dosage control, effect monitoring and other issues also need to be considered to ensure food safety and quality. For chitosan coatings, salicylic acid immersion, heat treatment, *etc.*, combined with other preservative double treatments are better and can be quickly applied in large-scale postharvest production lines. However, some factors still need to be considered in the actual promotion process, such as the standardization of technical operation, the investment cost of equipment, and the impact on the environment and human health. The feasibility, economy and environmental friendliness of the technology need to be considered comprehensively when promoting its application to ensure its effectiveness and sustainability. For genetic engineering, relatively few studies have been conducted to extend a certain index of the postharvest storage quality of melons by controlling the expression of specific genes in melons. Although genetic engineering technology has such potential in theory, it still faces some challenges in practical applications. Different regions, different seasons and different varieties of melons may have different responses to preservatives, which requires more experimental validation, and more large-sample, long-cycle experiments as well as comprehensive consideration of various factors are still needed to form a more complete and diversified system of fruit storage and control methods.

Research has focused mainly on observing the storage quality of fruit, determining the physiological and biochemical indices of fruit, determining cell membrane permeability and determining aroma volatiles. All the research content remains in the theoretical stage, and production practice application ability is still lacking. Different regions, different seasons

and different varieties of melons may have different responses to preservatives, which requires more experimental validation, and more large-sample, long-cycle experiments as well as comprehensive consideration of various factors are still needed to form a more complete and diversified system of fruit storage and control methods. Improving melon quality in the future and reducing the economic losses caused by storage and transportation are vital for the postharvest storage of melons and provide a theoretical reference for the prosperous development of the melon industry.

## ACKNOWLEDGEMENTS

Thanks to the support of Xinjiang Special Melon and Fruit Variety Improvement and Logistics Transportation Joint Research Center of Horticulture College of Xinjiang Agricultural University.

### Funding

This work was supported by the National Natural Science Foundation of China (No. 32360761), the Natural Science Foundation of Xinjiang Uygur Autonomous Region (No. 2022D01B27), the Key Research and Development Program of Xinjiang Uygur Autonomous Region (No. 2023B02017), the earmarked fund for XJARS (No. XJARS-06), the Natural Science Foundation of Hebei Province (C2022408017), the Science and Technology Project of Hebei Education Department (QN2022174). The Xinjiang Special Melon and Fruit Variety Improvement and Logistics Transportation Joint Research Center of Horticulture College of Xinjiang Agricultural University provided financial support and the use of part of the test platform. The funders had no role in study design, data collection and analysis, decision to publish, or preparation of the manuscript.

### Grant Disclosures

The following grant information was disclosed by the authors:
National Natural Science Foundation of China: 32360761.
Natural Science Foundation of Xinjiang Uygur Autonomous Region: 2022D01B27.
Key Research and Development Program of Xinjiang Uygur Autonomous Region: 2023B02017.
XJARS: XJARS-06.
Natural Science Foundation of Hebei Province: C2022408017.
Science and Technology Project of Hebei Education Department: QN2022174.
Xinjiang Special Melon.
Fruit Variety Improvement and Logistics Transportation Joint Research Center of Horticulture College of Xinjiang Agricultural University.

### Competing Interests

The authors declare there are no competing interests.

## Author Contributions

- Haofei Wang performed the experiments, prepared figures and/or tables, authored or reviewed drafts of the article, and approved the final draft.
- Jiayi Cui analyzed the data, authored or reviewed drafts of the article, and approved the final draft.
- Rui Bao performed the experiments, prepared figures and/or tables, and approved the final draft.
- Hui Zhang analyzed the data, prepared figures and/or tables, and approved the final draft.
- Zi Zhao analyzed the data, prepared figures and/or tables, and approved the final draft.
- Xuanye Chen performed the experiments, authored or reviewed drafts of the article, and approved the final draft.
- Zhangfei Wu analyzed the data, authored or reviewed drafts of the article, and approved the final draft.
- Chaonan Wang conceived and designed the experiments, authored or reviewed drafts of the article, and approved the final draft.

## Data Availability

This is a literature review.

## Supplemental Information

Supplemental information for this article can be found online at http://dx.doi.org/10.7717/peerj.17800#supplemental-information.

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
