# Peer review of "Research progress on the effects of postharvest storage methods on melon quality"

_PeerJ, doi:10.7717/peerj.17800_

## Round 0.1 · original submission · Major Revisions

· Academic Editor

Major Revisions

I request the author to address the comments from the reviewers.

Reviewer 1 ·

Basic reporting

The topic was not clear. The authors simply listed some related articles and did not provide in-depth discussion.

Experimental design

Review article.

Validity of the findings

No in-depth discussion. Simply listed related articles together.

Additional comments

lack of novelty and depth

Reviewer 2 ·

Basic reporting

a. The manuscript is generally well-written. However, there are minor issues with grammar and phrasing that could be streamlined for better readability. For example, some sentences are overly complex or contain redundant words, which can obscure the intended meaning and make the text less accessible. Spelling errors in English words also need to be corrected.

b. The manuscript extensively cites current literature and provides a thorough background on the postharvest storage of melons. It covers various methods and their impacts on melon quality. The manuscript covers various methods and their impact on the quality of melons, providing a comprehensive summary.

c. The structure of the manuscript adheres to academic standards with clear sections including the introduction, methods, results, and conclusions. Figures are used effectively to support the findings, but some minor modifications are necessary.

Experimental design

a. The content is appropriate for the journal's focus on biological science.

b. The manuscript complies with technical and ethical standards.

c. For a review manuscript, the methodology of literature collection and analysis is adequately detailed.

Validity of the findings

a. The topic is relevant to current agricultural practices and offers practical insights that can help improve melon storage and extend its shelf life, which is crucial for reducing waste and enhancing food security.

b. The conclusions are directly tied to the evidence reviewed, summarizing the effectiveness of different storage methods.

c. The manuscript largely summarizes existing literature without much critical analysis or discussion of the limitations of the reviewed studies. In the "Conclusion" section, it is suggested to include which method from the existing research is more advantageous for melon postharvest storage, and to discuss the limitations of the studies reviewed, which could provide a more comprehensive and balanced perspective.

Additional comments

Figures:
1. In Figure 1, the word "layer" appears twice in "outer layer".

2. Figure 2 legend, the word "figure" is misspelled in the sentence "The following abbreviations are used in the fgure."

3. Figure 3, since the author explains the process of reactive oxygen species (ROS) scavenging, please complete the diagram accordingly. It is recommended to organize the diagram based on antioxidant enzymes (SOD, APX, CAT, GPX, etc.) and non-enzymatic antioxidants (ascorbic acid, glutathione, flavonoids, etc.).
In the sentence "Superoxide dismutase is the only enzyme present...", avoid using absolute terms like "only" unless supported by sufficient references.

Main text:
1. As an academic publication, it is recommended to remove references to the use of Sci-Hub for searches throughout the text.

Reviewer 3 ·

Basic reporting

no comment

Experimental design

no comment

Validity of the findings

no comment

Additional comments

The paper was able to describe more comprehensively the research progress on the effects of postharvest storage methods on the quality of melon, but the linguistic presentation of individual statements was not sufficiently scholarly and professional and needs to be polished.

---

## Round 0.2 · accepted · Accept

· Academic Editor

Accept

Accepted and ready to go for production